# Stroke risk associated with NSAIDs uses in women with dysmenorrhea: A population-based cohort study

**Ya-Wen Lin[1], Jong-Yi Wang[2]ᵒ, Ming-Hung Lin [3,4]ᵒ \***

**1** School of Nursing and Graduate Institute of Nursing, China Medical University, Taichung, Taiwan,
**2** Department of Health Services Administration, China Medical University, Taichung, Taiwan, **3** Department of Pharmacy, Chia Nan University of Pharmacy and Science, Tainan, Taiwan, **4** Department of Nursing, National Taichung University of Science and Technology, Taichung, Taiwan

ᵒ These authors contributed equally to this work.
\* lmh.roger@msa.hinet.net

## Abstract

### Objective

Dysmenorrhea is among the most common type of gynecological problem, affecting young women across the globe. This study assessed the comparative risk of stroke associated with the use of nonsteroidal anti-inflammatory drugs (NSAIDs) and non-NSAIDs in women with dysmenorrhea while taking into account the following factors such as age, history of pregnancy, NSAIDs uses and its duration of use, and selected comorbidities.

### Methodology

We used a quantitative research approach based on a comparative case-control study design. The study data was selected from the Longitudinal Health Insurance Database (LHID) 2000, of the Taiwan National Health Research Institutes. Among the estimated 23.4 million insured Taiwanese, who were covered by the Taiwan health insurance system, in the 2000 registry of beneficiaries, one million individuals were randomly selected from the database. A total of 24,955 females suffering from dysmenorrhea were selected for the study. Out of those 3238 (13%) participated in the study group and 21,717 (87%) were randomly distributed into the controls group. Women in the age range, 15–49 years, who did not have any history of stroke, hysterectomy, and/or ovariectomy, were included in the study. A comparative proportional distribution analysis was used for data analysis.

### Results

Age and use of NSAIDs and its duration of usage were factors associated with an increased incidence of stroke. The stroke incidence rate was 12.77 per 10,000 person-years, and 1.83-fold higher in NSAIDs use cohort than in comparisons with adjusted hazard ratio (aHR) of 1.47 (95% CI = 0.93–2.32). Among women with dysmenorrhea use of NSAIDs, the stroke incidence increased to 2.29-fold (aHR 95% CI = 1.36–3.84) in those use for ≧24 days per month and to 0.51-fold (aHR 95% CI = 0.13–2.10) in those use for 6–12 days per month.

**Data Availability Statement:** The database was obtained from the Taiwan National Health Research Institutes. We are not eligible to duplicate and disseminate the database. For further access to the database, please contact the Ministry of Health and

Welfare (Email: stcarolwu@mohw.gov.tw) for assistance. Taiwan Ministry of Health and Welfare Address: No.488, Sec. 6, Zhongxiao E. Rd., Nangang Dist., Taipei City 115, Taiwan (R.O.C.). Phone: +886-2-8590-6848).

**Funding:** This study was supported in part by grants from the Ministry of Health and Welfare, Taiwan(MOHW108-TDU-B-212-133004), China Medical University Hospital (DMR-108-024 and DMR-108-111), Ministry of Science and Technology, Taiwan (Grant no. MOST108-2410-H-039-001, MOST109-2410-H-039-001, and MOST110-2410-H-039-001) and China Medical University (Grant no. CMU109-MF-119 and CMU110-MF-123). No additional external funding was received for this study.

**Competing interests:** The authors have declared that no competing interests exist.

## Conclusions

Women with dysmenorrhea who use NSAIDs have a higher risk of stroke. Especially young women, the risk of stroke is further increased, and the longer the medication, the higher the risk of stroke. Every woman with symptoms of dysmenorrhea deserves specialized outpatient treatment and care.

## Introduction

Dysmenorrhea is among the most common types of gynecological problem affecting young women across the globe [1–5]. Although there are notable variations across different communities around the world, the prevalence of dysmenorrhea ranges from 50–90% [1]. It is usually characterized by aching abdominal pain, present before the onset, and during menstruation, which is generally the most common symptom seen in women. Nausea tiredness, headache, itchiness, diarrhea, and vomiting are some other accompanying symptoms [1, 3]. Dysmenorrhea can be of two types: primary dysmenorrhea (PD) or secondary dysmenorrhea (SD). PD is the type of dysmenorrhea that manifests itself without any accompanying organic pathological condition in the reproductive organs, while SD is caused by an organic pathological condition in the reproductive system [4]. In addition, PD differs from SD in that, its occurrence usually starts shortly after menarche while that of the former is dependent on the presence or absence of an organic disease(s) in the reproductive system [1, 4].

There are a variety of factors that have been closely linked with the risk of developing dysmenorrhea. The key contributing factor is an increase in the secretion level of hormones, mainly vasopressin and oxytocin [6]. Notably, alterations in hormonal balance is not only common among women who experience abnormal menstrual patterns, but such alterations are also a risk factor for mediating the pathogenesis of hypertension [5, 6]. It is noticed that women with dysmenorrhea are at a high risk of sustaining blood pressure and other related health complications [5]. This is a major health concern given the general prevalence of practicing the use of nonsteroidal anti-inflammatory drugs (NSAIDs)as the first-line therapy for the treatment of PD [2, 3, 6]. Moreover, owing to its high prevalence, dysmenorrhea is frequently considered as a self-treated health condition among patients of all age groups. Specifically, teenagers are prone to experiencing PD-related symptoms, with an estimated 60% of all incidences of PD being reported during the teenage years [7].

Despite being relatively premature, studies have shown that over 84% of teenagers who experience PD engage in self-treatment of associated symptoms, without seeking any advice from a physician and about 78% of them choose medications without adult supervision [7, 8]. In fact, studies have reported that not only most of these teenagers choose their medications based on convenience and availability [2] but also over half of them lack understanding of the correct dosing requirements of the over the counter (OTC) medications used for self-treatment of PD [7, 8]. Furthermore, even among adults, incidences of polypharmacy and incorrect dosages and/or dosing frequency of taking NSAIDs for self-treatment of PD-related symptoms is not uncommon [2, 8]. Unfortunately, this trend of self-treatment among women dealing with PD is continuing, despite recent research findings which have associated long-term use of NSAIDs with increased risk for undesirable cardiovascular outcomes among patients [6].

Particularly, it is seen that extended use of NSAIDs among women with dysmenorrhea, also increases the risks for of sustaining common cardiac adverse effects such as hypertension, stroke, bleeding and heart attack [8–13]. However, despite this consensus on the association of

long-term use of NSAIDs with increased risk of adverse cardiovascular effects, there still remains an inherent lack of unanimity on whether or not women with dysmenorrhea are at a higher risk of sustaining stroke from the blood pressure elevation, caused by an alteration in the level of their hormones due to extended use of NSAIDs. The risk of sustaining a stroke excessively exacerbates the detrimental effects of dysmenorrhea on women, impacting not only them individually, but also their family members and the society as a whole. It needs to be emphasized that stroke is the third leading cause of death across the globe and is a common cause of disability and mortality in societies around the world [13, 14].

Additionally, there are notable variations not only in terms of potential stroke risk but also on the type of stroke detected among women who have and those who do not have dysmenorrhea [6]. In spite of the widespread prevalence of stroke risk in women with dysmenorrhea, there are no detailed clinical studies based on the relationship of dysmenorrhea, extended use of NSAIDs, and stroke risk as a health concern. In this study, we sought to investigate the comparative risk of stroke among women affected with dysmenorrhea, associated with NSAIDs use and non-NSAIDs use. We conducted a case-control study to assess stroke risk in such women by taking into account various factors such as age, history of pregnancy, NSAIDs use, and duration of use. Our aim was to identify the role of selected comorbidities in NSAID users. Data for this study was obtained from insurance claims of the Taiwan National Health Research Institutes database, by selecting women who suffered from dysmenorrhea and were prescribed NSAIDs at least two times.

## Methodology

### Study design and study population

In this study, we used a quantitative research approach based on a comparative cross-sectional case-control study design. The choice of conducting a case-control study design for this study was essential to ensure that both the cases and controls were sampled from an existing source population, that was clearly predefined in relation to details of its sample size [16]. Specifically, this study design offers the benefit of eliminating the problem of referral or partial verification bias, commonly seen with typical case-control designs. Moreover, this study design also ensured that all the patients who were selected for the study signify a true representation of the actual disease status and act as a standard reference of interest for the study [15, 16]. Furthermore, a nested case-control design allows a simple way for obtaining all the measures of diagnostic accuracy for the target sample [16]. The study data was selected from the Longitudinal Health Insurance Database (LHID) 2000, of the Taiwan National Health Research Institutes. Out of the estimated 23.4 million insured Taiwanese who were covered by the Taiwan health insurance system, as documented in the 2000 registry of beneficiaries, one million individuals were randomly selected from the database. As a subset of the insurance claims data of the registry of beneficiaries, LHID 2000 is a reliable representation of the entire population of Taiwan. Therefore, to conduct this study, this database was chosen as the study population. It has been reported that from 1997, coverage of the residents of Taiwan by the national health insurance system has been estimated to be more than 99% of the entire population [17]. Additionally, the LHID 2000 database of the Taiwan National Health Research Institute boasts about maintaining comprehensive documentation of all insurance claims data of the involved beneficiaries, including their specifics such as demographics, medications prescriptions, surgical procedures, and clinical examinations, as well as information regarding inpatient and outpatient care offered and cost of care involved, from 1996 to 2013. While conducting the study, permission to access and use the insurance claims data from the database was approved by the Research Ethics Committee of China Medical University and Hospital, Taichung, Taiwan

(CMUH104-REC2-115). During our study research, we adhered to the principles as mentioned in the Declaration of Helsinki.

## Selection criteria

From the claims data in the LHID 2000 database, we identified 27,749 females with diagnosed dysmenorrhea (ICD-9-CM: 625.3) for at least twice from 2000 to 2008. After excluding those aged < 15 or > 50 years (n = 1545) and those with the history of hysterectomy or ovariectomy (Procedure Code 68 and 65, respectively), or stroke (n = 1249). The dysmenorrhea diagnosed date was defined as index date (Fig 1). Of the remaining 24,955 women with dysmenorrhea were eligible for this study and divided into two groups. The comparison group included 21,717 women who did not have any documented history of using NSAIDs or had used NSAIDs only once, for the treatment of dysmenorrhea, since the time they were diagnosed with the condition. Seventy people in the control group had a stroke. The control group included 3238 with documented of using NSAIDs to the treatments of dysmenorrhea, at least twice and 27 people had strokes.

## Stroke outcome and associated factors

For each woman, we checked whether they had a stroke (ICD-9 code:430–437) [6, 11]. Stroke types include hemorrhagic stroke (HS, ICD-9 codes 430–432), ischemic stroke (IS, ICD-9 code 434), transient cerebral ischemia (TIA, ICD-9 code 435) and other acute and other ill-defined cerebrovascular diseases (OCD, ICD-9 code 436 and 437). The demographic data file provided information on age was stratified into five subgroups (15–19, 20–24, 25–29, 30–39 and 40–49 years), and pregnancy (ever and no). We also searched for comorbidities that were potentially linked to stroke including: diabetes (ICD-9 code: 250), hyperlipidemia (ICD-9 code: 272), obesity (ICD-9 code: 278), hypertension (ICD-9 code: 401–405), thyroid disease (ICD-9 code: 420–426), arrhythmia (ICD-9 code: 427) and alcohol-related diseases (ICD-9 code: 291, 3030, 305, 571.0–571.3, 790.3, V11.3) were specifically examined among those dysmenorrhea women. All baseline comorbidities were defined before the index date.

## Data analysis

We used SAS software version 9.4 (SAS Institute Inc, Carey USA) to perform the data analyses for this study, with a p-value of <0.05 considered statistically significant. Chi-square test was conducted to compare the difference of categorical variables between cohorts with and without NSAIDs use. Data analysis involved the use of logistic regression analysis as the statistical method. Baseline demographic statuses and comorbidities were compared between the cases and controls. Logistic regression analysis was used to calculate the crude and adjusted odds ratios (cOR and aOR) of stroke and the 95% confidence intervals (CI) to identify whether these variables were also associated with NSAIDs user. We used a multivariable-adjusted model to estimate the average risk of stroke in women dealing with dysmenorrhea, who used NSAIDs and in those who did not use NSAIDs. The NSAIDs cohort to the comparison cohort hazard ratio (HR) and 95% confidence intervals (CI) were assessed using Cox proportional hazards regression analysis. The multivariable Cox model was used to estimated adjusted hazard ratio (aHR), controlling for age, pregnancy and all comorbidities. A P-value of 0.05 was used as the statistical significance. To determine whether the risk of stroke is associated with the length of time NSAIDs being used and also verifying its association, a multivariable-adjusted model was used to estimate the average stroke risk and proportional distributions of stroke risk, by calculating the number of days per month a woman with dysmenorrhea women with dysmenorrhea used NSAIDs. The duration for which NSAIDs was used by women with

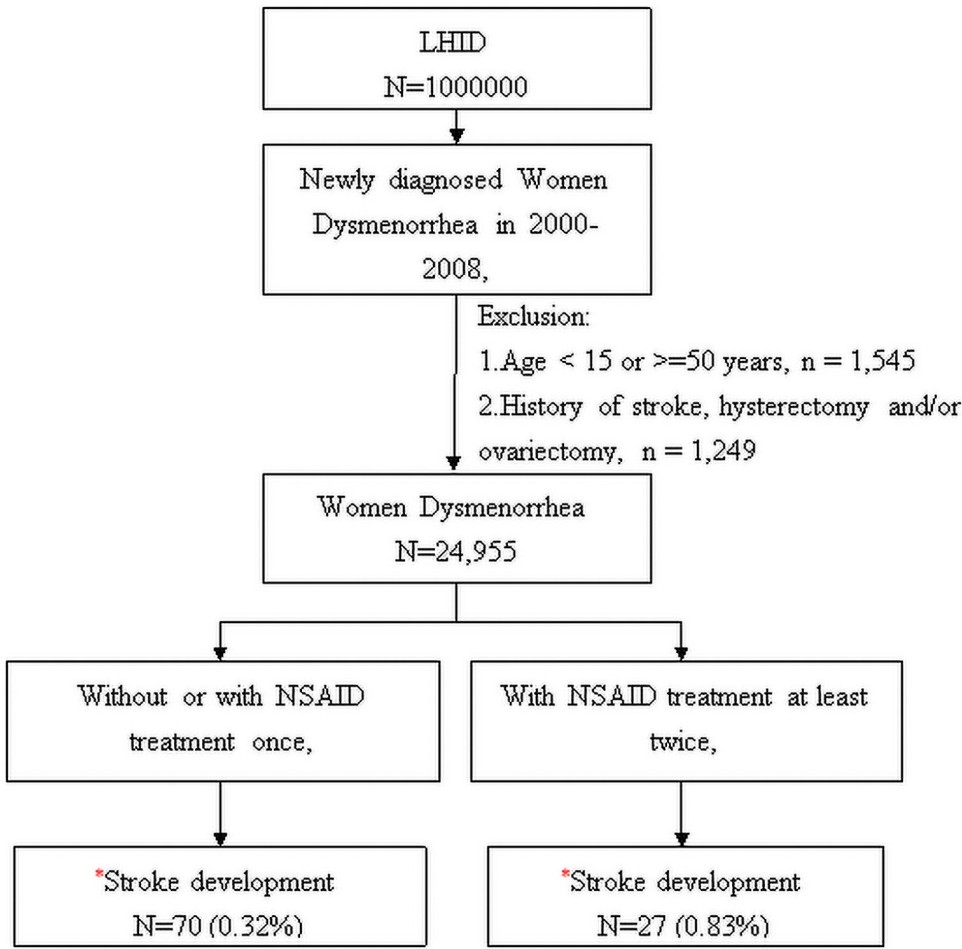

**Fig 1. Flowchart showing the process of sample selection.** *p<0.001.

dysmenorrhea was stratified into five subgroups: none, ≤ 5, 6–12, 13–24, and ≥ 24 days per month. The model was adjusted for age and pregnancy. The significance level was examined by conducting the t-test. We used Kaplan-Meier method to calculate and plot the cumulative incidence of stroke for both cohorts, with the difference examined by log-rank test. The incidence of stroke was calculated by dividing the sum of stroke development by the sum of follow-up person-years, presented by 100 person-years.

## Results

For this study, a total of 27,749 females were identified from the LHID 2000 database as newly diagnosed dysmenorrhea cases, who have filed insurance claims from 2000 to 2008. After excluding females who were of age, below 12 years and 50 years and above (n = 1545) and those who had known history of stroke, hysterectomy, and/or ovariectomy (n = 1249) by the time they were first diagnosed with dysmenorrhea, the study included a total of 24,955 remaining adolescents and women, of age group15-49 years. The study comprised of 3238 (13%) NSAID treatment cases (study group) and 21717 (87%) non-NSAID treatment patients (control group) (Fig 1).

**Table 1. Baseline comparison in women with dysmenorrhea, with and without NSAIDs uses.**

| | NSAID users | | | | p-value | Odds ratio(95%CI) | |
| --- | --- | --- | --- | --- | --- | --- | --- |
| | Yes | | No | | | | |
| | N = 3238 | | N = 21717 | | | | |
| | (13.0%) | | (87.0%) | | | | |
| Age, years | n | (%) | n | (%) | 0.009 | Crude | Adjusted |
| 15–19 | 345 | (10.7) | 2703 | (12.5) | | Ref. | |
| 20–24 | 690 | (21.3) | 4508 | (20.8) | | 1.20 (1.05–1.38)** | 1.15 (1.00–1.32) |
| 25–29 | 726 | (22.4) | 4950 | (22.8) | | 1.15 (1.00–1.32)* | 1.03 (0.90–1.19) |
| 30–39 | 945 | (29.2) | 6360 | (29.3) | | 1.16 (1.02–1.33)* | 0.96 (0.83–1.10) |
| 40–49 | 532 | (16.4) | 3196 | (14.7) | | 1.30 (1.13–1.51)*** | 1.01 (0.87–1.18) |
| Mean (SD) | 30.2 | (8.58) | 29.7 | (8.57) | 0.001 | | 1.00 (0.99–1.00) |
| Follow-up years | | | | | | | |
| Mean (SD) | 6.53 | (3.06) | 4.75 | (3.08) | <0.0001 | | |
| Pregnancy | | | | | <0.0001 | | |
| Ever | 1386 | (42.8) | 8336 | (38.4) | | 1.20 (1.12–1.30)*** | 1.17 (1.08–1.27)*** |
| No | 1852 | (57.2) | 13381 | (61.6) | | Ref. | Ref. |
| Comorbidity | | | | | | | |
| Diabetes | 123 | (3.80) | 493 | (2.27) | <0.0001 | 1.70 (1.39–2.08)*** | 1.26 (1.01–1.57)* |
| Hypertension | 144 | (4.45) | 605 | (2.79) | <0.0001 | 1.62 (1.35–1.96)*** | 1.30 (1.06–1.58)* |
| Hyperlipidemia | 218 | (6.73) | 895 | (4.12) | <0.0001 | 1.68 (1.44–1.96)*** | 1.36 (1.15–1.62)*** |
| Obesity | 41 | (1.27) | 160 | (0.74) | 0.002 | 1.73 (1.22–2.44)** | 1.36 (0.95–1.93) |
| Arrhythmia | 229 | (7.07) | 953 | (4.39) | <0.0001 | 1.66 (1.43–1.93)*** | 1.46 (1.26–1.71)*** |
| Thyroid disease | 416 | (12.9) | 1988 | (9.15) | <0.0001 | 1.46 (1.31–1.64)*** | 1.32 (1.18–1.48)*** |
| Alcohol-related diseases | 48 | (1.48) | 165 | (0.76) | <0.0001 | 1.97 (1.42–2.72)*** | 1.76 (1.27–2.45)*** |

NSAIDs: Nonsteroidal anti-inflammatory drugs.

Adjusted OR: multiple analysis after adjusting for age, pregnancy, comorbidities; CI, confidence interval.

* p <0.05,

** p <0.01,

*** p <0.001.

The mean ages for the cases and the controls were 30.2 (SD = 8.58) and 29.2(SD = 8.57) years, respectively (Table 1). The lack of a history of pregnancy increased the risk of being diagnosed with dysmenorrhea among women in the country (57.2% vs 42.8% in the cases and 61.6% vs. 38.4% in the controls). The results showed that females of age group, 30–39 years had the highest number of newly diagnosed dysmenorrhea (n = 945 (29.2%) of cases and n = 6360 (29.3%) of controls), while those between12-19 years had the least number (n = 345 (10.7%) of cases and n = 2703 (12.5%) of controls) (Table 1). In terms of the prevalence of stroke risk associated with age in women with dysmenorrhea, the study showed that out of the total sample, 70 (0.32% of the total study sample) and 27 of the women (0.83% of the total sample for the study) developed stroke in the controls and cases groups, respectively (Fig 1).

The results highlighted that the probability for developing stroke increases with age as seen among the cases and controls group overall (n = 29 at 40–49 years vs. n = 3 at 15–19 years in the controls and n = 7 at 40–49 years vs. n = 2 at 15–19 years in the cases) (Table 2). However, compared to the control group, the risk of stroke was the highest in the study group (cases) among women of age between 30–39 years (n = 24, CI = 95%). This is a significant finding as this age group accounted for more than half of the total number that developed stroke among the case group. It was seen that incidence rate per 10000 person- was 12.77 in the cases group

**Table 2. Risk of stroke compared between NSAIDs use cohort and comparison cohort among women with dysmenorrhea.**

| Age, years | Cohort | Event | PY | IR | HR (95% CI) | | | |
|---|---|---|---|---|---|---|---|---|
| | | | | | Crude | p | Adjusted* | p |
| 15–19 | Comparison | 3† | 16667 | 1.80 | 1.0 | | 1.0 | |
| | NSAIDs | 2† | 2581 | 7.75 | 4.81 (0.80–28.8) | 0.09 | 5.23 (0.87–31.3) | 0.07 |
| 20–24 | Comparison | 5 | 23965 | 2.09 | 1.0 | | 1.0 | |
| | NSAIDs | 4† | 4749 | 8.42 | 3.95 (1.05–14.8) | 0.04 | 4.27 (1.13–16.1) | 0.03 |
| 25–29 | Comparison | 8 | 22186 | 3.61 | 1.0 | | 1.0 | |
| | NSAIDs | 0 | 4727 | 0.00 | NA | | NA | |
| 30–39 | Comparison | 25 | 26731 | 9.35 | 1.0 | | 1.0 | |
| | NSAIDs | 14 | 5754 | 24.33 | 2.46 (1.27–4.76) | 0.008 | 2.22 (1.13–4.36) | 0.02 |
| 40–49 | Comparison | 29 | 13510 | 21.47 | 1.0 | | 1.0 | |
| | NSAIDs | 7 | 3327 | 21.04 | 0.91 (0.40–2.10) | 0.83 | 0.70 (0.30–1.62) | 0.40 |
| All | Comparison | 70 | 103058 | 6.79 | 1.0 | | 1.0 | |
| | NSAIDs | 27 | 21138 | 12.77 | 1.83 (1.17–2.86) | 0.008 | 1.47 (0.93–2.32) | 0.10 |

PY: person-years; IR: incidence rate per 10000 PY; HR: hazard ratio; CI: confidence interval.

*Model was adjusted for age, pregnancy and all comorbidities listed in Table 1.

† event: less than 5.

compared to 6.79 in the control group. Notably, it was evident in the study that women who were suffering from dysmenorrhea and had a history of using NSAID for 24 days or more, per month, not only contributed to majority of the cases, but were also at the highest risk for developing stroke (n = 20, out of the 27 women who developed stroke in the cases group) (Table 3).

The results showed that in general, the risk of developing stroke increases with increase in the number of days per month that a woman who has dysmenorrhea uses NSAID as a treatment option (IR = 24.76 for cases who used NSAID treatment for 24 or more days per month) (Table 3). The study results did not show any evidence linking, NSAID for treatments use at the rate of 5 or less days per month to any risk of developing stroke among women with dysmenorrhea, regardless of their age, pregnancy and comorbidity status. However, the data indicated an overall higher risk of developing stroke among women with dysmenorrhea, who had one or more comorbidity and were using NSAIDs for the treatment.

Table 4 indicates that the stroke risk in the subgroup of women who had one or more comorbidity was higher than that posed by age (cSHR = 1.83 at p = 0.008 v. aSHR = 1.47 at

**Table 3. Stroke risk seen in women with dysmenorrhea in association with use of NSAIDs and its duration of usage.**

| | N | Event | PY | IR | HR (95% CI) | | | |
|---|---|---|---|---|---|---|---|---|
| | | | | | Crude | p | Adjusted* | p |
| NSAIDs use day, per month | | | | | | | | |
| None | 21717 | 70 | 103058 | 6.79 | 1.00 | | 1.00 | |
| < = 5 | 230 | 0 | 2006 | 0.00 | NA | | NA | |
| 6–12 | 763 | 2 † | 5472 | 3.65 | 0.52 (0.13–2.13) | 0.36 | 0.51 (0.13–2.10) | 0.35 |
| 13–24 | 851 | 5 † | 5581 | 8.96 | 1.30 (0.52–3.21) | 0.58 | 1.17 (0.47–2.91) | 0.73 |
| ≧24 | 1394 | 20 | 8079 | 24.76 | 3.56 (2.16–5.86) | <0.0001 | 2.29 (1.36–3.84) | 0.002 |

PY: person-years; IR: incidence rate per 10000 PY; HR: hazard ratio; CI: confidence interval.

*Model was adjusted for age, pregnancy and all comorbidities listed in Table 1.

† event: less than 5.

**Table 4. Subhazard ratios of stroke based on the comparison linked-risk regression.**

|  | Comparison group | NSAIDs | p |
| --- | --- | --- | --- |
| Stroke |  |  |  |
| cSHR (95% CI) | 1.0 | 1.83 (1.17–2.88) | 0.008 |
| aSHR* (95% CI) | 1.0 | 1.47 (0.92–2.36) | 0.11 |

SHR, subhazard ratio; CI: confidence interval.

*Model was adjusted for age, pregnancyand all comorbidities and medications listed in Table 1.

0.11). Notably, as evident in Table 1, hyperlipidemia (ICD-9 code: 272), thyroid disease (ICD-9 code: 420–426), and arrhythmia (ICD-9 code: 427) were the most common, while obesity (ICD-9 code: 278) and alcohol-related diseases (ICD-9 code: 291, 3030, 305, 571.0–571.3, 790.3, V11.3) were the least common types of comorbidities seen among the cases in this study. In addition, we found that diabetes (ICD-9 code: 250) and hypertension (ICD-9 code: 401–405) were moderately prevalent among women who suffered from dysmenorrhea, in both the cases and controls groups of the study.

Fig 2 shows that the number of cumulative incidences of stroke in women who had dysmenorrhea and used NSAID treatments, was significantly higher compared to those who did not use NSAID treatment over the follow-up time. Moreover, this effect was higher in the case group as compared to that of controls, but there was no significant difference noted with regard to the timing of developing stroke between both the groups. In both the groups, the risk of stroke peaked and relatively flattened, after 12 months of follow-up time (CI = 0.02 for cases and 0.01 for controls at p = 0.07).

## Discussion

The study results indicated that generally, there is a significant of using NSAID as a treatment among women who had newly diagnosed dysmenorrhea have a higher risk of stroke. Especially young women, the risk of stroke is further increased, and the longer the medication, the higher the risk of stroke. Previous research studies that were conducted on the risk of developing stroke in women with dysmenorrhea have attributed an increased risk of stroke, to alterations in hormonal balance which in turn causes incidents of pathogenesis of hypertension [5, 6]. In particular, hormonal imbalances and hypertension are not only common among women who experience abnormal menstrual patterns, but also that these conditions are associated with increasing the risk of sustaining blood pressure and related health complications, including stroke in women with dysmenorrhea [5]. The present study explored the comparative risk of developing stroke, associated with NSAID treatment, and non-NSAID treatment use among women with dysmenorrhea with or without the comorbidities, which have been generally linked to increase the likelihood of causing stroke. There were fewer incidents of stroke reported among women in the controls group as compared to the cases group. This study demonstrated more incidents of stroke among women in the cases group who used more NSAID-based treatments per month as compared to those who used fewer NSAIDs per month.

This study showed that particularly, women who had dysmenorrhea and used NSAID treatments were twice as likely to develop stroke as compared to women who did not use NSAID treatments. This indicates that use of NSAIDs can significantly increase the risk of developing stroke in women with dysmenorrhea. However, this risk increases with the rate of usage of NSAIDs per month. The results demonstrated that women who used NSAIDs for 24 days or

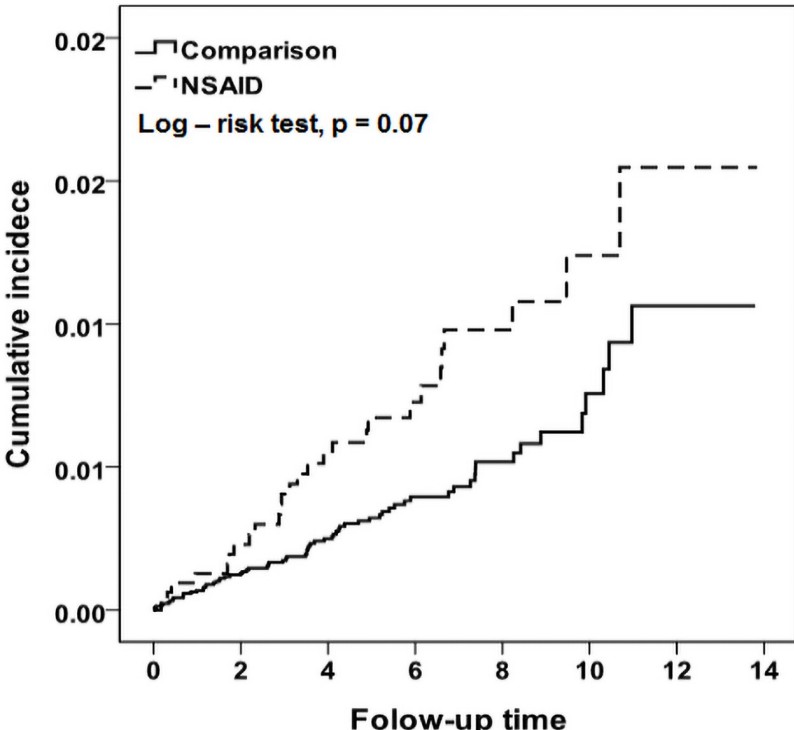

**Fig 2. Log-risk test for evaluating the comparative cumulative incidence of stroke among NSAID users and non-NSAID users.**

more in a month were nearly 25 times more likely to develop stroke than those who engaged in occasional NSAIDs usage. The stroke risk in women with dysmenorrhea also increased with age and in presence of comorbidities.

It is worth noticing that unlike previous research where prevalence of newly diagnosed dysmenorrhea was comparatively high among women in their teenage and early adulthood, this study found that the condition was more prevalent among age group, 20–39 years (72.9%) in both case and control group. This disparity can be attributed to the fact that, the present study did not adjust the data analysis models according to the differences in varied types of dysmenorrhea that were being investigated. This implies that the high prevalence of dysmenorrhea found among women of age group, 20–39 years, could be due to combination effects of both PD and SD. The high prevalence of self-treatment of dysmenorrhea among patients, regardless of their age, can also account for the small numbers of adolescents and young adults aged below 20 years who sought related treatment interventions from formal health care services in the nation over the studied period. Given the proximity and inherent sharing of cultural values, tastes, and preferences between Taiwanese and Chinese population, the usage of traditional Chinese medicine for dysmenorrhea treatment remains high among women of Taiwan. Hence, in Taiwan this further prevents the utilization of formal health care services among women with dysmenorrhea.

It is reported that both self-preparation and self-administration of traditional Chinese medicines for treating dysmenorrhea are common practices among women with dysmenorrhea. Moreover, previous studies that have compared the clinical efficacy of self-prepared dysmenorrhea paste, in combination with other Chinese herbal medicine practices such as umbilical therapy and acupuncture, have shown relatively higher effective rate for treating PD

as compared to studies where conventional medicines were used [18–20]. Literature also showed that high utilization of traditional Chinese medicine as compared to conventional western drugs, in China and its neighboring countries such as Taiwan, is generally due to a perception that a western medicine in the long-term are associated with higher risks, impacting the overall health and wellbeing of users [1, 21–23]. Overall, results of the study demonstrated that the risk of developing stroke among women with dysmenorrhea is associated with various other factors such as age, NSAIDs usage frequency, any prior history of pregnancy and/or any other comorbidity which have been closely linked to stroke risk.

It was seen that the proportion of women with dysmenorrhea who used NSAIDs and developed stroke was comparatively greater than that of women who did not use NSAIDs and developed stroke. In the study, incidence rate was 12.77 in the case group as compared to 6.79 in the control group. This indicates that there is a two-fold increase in the likelihood of developing stroke among women who have used NSAIDs for the treatment of dysmenorrhea. The findings in the present study have attributed these common comorbidities such as hyperlipidemia (ICD-9 code: 272), thyroid disease (ICD-9 code: 420–426) and arrhythmia (ICD-9 code: 427) with an increased risk of developing stroke among women with dysmenorrhea which is consistent with results shown in previous research where hypertension and hormonal imbalances were linked with increased stroke risk [6, 24, 25]. These comorbidities are one of the major risk factors for initiating stroke among women with dysmenorrhea, who generally prefer to use NSAID as a treatment option.

As previously discussed that the extended use of NSAIDs among women with dysmenorrhea has been linked with an increasing risks for sustaining common cardiovascular adverse effects such as hypertension, stroke, bleeding and heart attack [8–13]. However, more importantly, the risk of developing NSAID-related adverse cardiovascular outcomes among patients, increases with polypharmacy or the use of multiple NSAIDs, commonly seen in patients who have multiple chronic comorbidities. It needs to be emphasized that stroke is the third leading cause of death across the globe and often leads to disability and mortality, affecting the population worldwide. This impact is mainly due to its association with various chronic comorbidities as seen in our modern society [13, 14, 23, 24]. In addition, the health risks associated with consumption of NSAIDs are largely cumulative in nature [25–29]. In other words, the risk of sustaining adverse health reactions and related outcomes from using NSAIDs will generally increase with the increase in duration and frequency of using such treatments.

In this study, we found that the cumulative incidence rates of stroke were higher in the cases than those of the controls. The present study also found that the risk of stroke is significantly increased with increase in the number of days per month that a woman with dysmenorrhea uses NSAID as a treatment modality 5(IR = 24.76 for cases who used NSAID treatment for 24 or more days per month) (Table 3). However, the study results did not show any evidence linking NSAIDs with treatments use at the rate of 5 or less days per month, to any risk of developing stroke in women with dysmenorrhea regardless of their age, pregnancy and comorbidity status. The findings of this study indicate that the risk of stroke associated with the use of NSAID among women with dysmenorrhea can be significantly blamed on the common practice of self-treatment for dysmenorrhea-related symptoms due to the convenience and cost-effective availability of NSAIDs as common general-use OTC medications in the society. However, with this lack of involvement of qualified healthcare professionals in the use of NSAIDs for self-treatment of dysmenorrhea-related symptoms presents with high likelihood of an 'as needed' attitude in the taking of these medications and associated risks of adverse drug outcomes, both in the short- and long-term, among these women.

Furthermore, given their preferential use as first-line medications of choice for pain treatment in practice, incidences of polypharmacy in taking NSAIDs are not uncommon. The

result of the present study indicated that the risk of stroke linked to use of NSAIDs, among women with dysmenorrhea differs, depending on the consumption of NSAIDs by individuals. This is quite likely given not only the inherent multiplicity of generics of NSAIDs available in the market, but also that the active elements in NSAIDs are common constituents in most other medications which are commonly used in clinical treatment. As such, discouraging the culture of self-treatment for dysmenorrhea by using OTC NSAIDs can significantly help in reducing the stroke risk associated with NSAIDs in women with dysmenorrhea. This approach will be conducive to improving the prospects of safe-prescribing of NSAIDs.

Overall, compared to the controls group, the prevalence of stroke among women with dysmenorrhea in the case group was notably high (0.32% vs. 0.83% in the non-NSAIDs and NSAIDs use groups, respectively). This finding seems to contradict previous studies which have found a general comparative clinical efficacy, if not better, for using traditional herbal medicine and practices compared to conventional medicines on PD [18–20]. The disparity can be explained by the fact that, just like the use of conventional medicines such as NSAIDs, use of traditional medicines continues to present with inadequacy of effectiveness and shows likely adverse outcomes due to the problems related to its self-preparation and treatment [19, 23]. This emphasizes the need for advocating professional aid in the treatment of dysmenorrhea as a critical measure, for reducing stroke risk among such women.

## Conclusions

In this study, we have identified the comparative stroke risk associated with the use of NSAIDs uses and non-NSAIDs uses among women dealing with dysmenorrhea and how age, pregnancy history and selected comorbidities impacts the overall risk. As a whole, from the present study it is clear that the risk of stroke in women using NSAIDs for dysmenorrhea, increases significantly with increase in the quantity and frequency of using NSAIDs. Additionally, comorbidities also increase the stroke risk in these women, as they are generally under the influence of multiple medications which further amplifies the risk. Therefore, this study supports previous research that recommends the need for discouraging self-treatment to deal with dysmenorrhea symptoms, in the absence of physician advice or lack of adult supervision.

## Limitation

Although this study was strengthened by the use of the large insurance claims database, there were several limitations. First, information on body mass index, lifestyle of diet, smoking and drinking, and family health history was unavailable to adjust for these potential confounders in data analyses. However, the impact from some of these factors might be minor because women are more likely to avoid unhealthy behaviors. Smoking and drinking are rare habits in women in Taiwan and obesity is not prevalent as well. The study results might not be generalizable to non-Chinese populations and populations with higher rates of obesity. Second, the issue of lacking the type of used NSAIDS, the dose of used NSAIDs, the time between NSAID consumption which might increase the stroke in these groups. Third, NSAIDs as common general use OTC medications in the Taiwan society and easy to buy at pharmacies. However, with this lack of involvement of qualified healthcare professionals (example; Doctor, Pharmacist) in the use of NSAIDs treatment of dysmenorrhea.

## Supporting information

**S1 Appendix. Disease ICD-9 code.**
(DOCX)

## Author Contributions

**Conceptualization:** Ya-Wen Lin, Ming-Hung Lin.

**Data curation:** Ya-Wen Lin, Ming-Hung Lin.

**Formal analysis:** Ya-Wen Lin.

**Investigation:** Ya-Wen Lin.

**Methodology:** Ya-Wen Lin.

**Project administration:** Jong-Yi Wang.

**Resources:** Jong-Yi Wang.

**Software:** Ya-Wen Lin, Ming-Hung Lin.

**Supervision:** Jong-Yi Wang, Ming-Hung Lin.

**Validation:** Ya-Wen Lin, Jong-Yi Wang, Ming-Hung Lin.

**Writing – original draft:** Ya-Wen Lin.

**Writing – review & editing:** Jong-Yi Wang, Ming-Hung Lin.

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
