## [Decision Letter · Decision Letter 0]

21 Jul 2021

PONE-D-21-13819

Stroke Risk associated with NSAIDs Uses in Women with Dysmenorrhea;a population-based Cohort Study

PLOS ONE

Dear Dr. Lin,

Thank you for submitting your manuscript to PLOS ONE. After careful consideration, we feel that it has merit but does not fully meet PLOS ONE’s publication criteria as it currently stands. Therefore, we invite you to submit a revised version of the manuscript that addresses the points raised during the review process.

Kindly address the reviewers comments and submit a point by point rebuttal letter. If you do not agree with reviewers comments, you can give your explanation with supporting evidence. Good luck

We look forward to receiving your revised manuscript.

Kind regards,

Muhammad Junaid Farrukh

Academic Editor

PLOS ONE

Journal Requirements:

2. Thank you for including your ethics statement:  "the database was approved by the Research Ethics Committee of China Medical University and Hospital, Taichung, Taiwan (CMUH104-REC2-115)".   

Please provide additional details regarding participant consent. In the ethics statement in the Methods and online submission information, please ensure that you have specified (1) whether consent was informed and (2) what type you obtained (for instance, written or verbal, and if verbal, how it was documented and witnessed). If your study included minors, state whether you obtained consent from parents or guardians. If the need for consent was waived by the ethics committee, please include this information.

"This study was supported in part by grants from the Ministry of Health and Welfare, Taiwan

(MOHW108-TDU-B-212-133004), China Medical University Hospital (DMR-108-024 and

DMR-108-111), Ministry of Science and Technology, Taiwan (Grant no. MOST108-2410-H-

039-001) and China Medical University (Grant no. CMU109-MF-119)"

"NO - Include this sentence at the end of your statement: The funders had no role in study design, data collection and analysis, decision to publish, or preparation of the manuscript"

5.Please amend your list of authors on the manuscript to ensure that each author is linked to an affiliation. Authors’ affiliations should reflect the institution where the work was done (if authors moved subsequently, you can also list the new affiliation stating “current affiliation:….” as necessary).

6. Please amend either the abstract on the online submission form (via Edit Submission) or the abstract in the manuscript so that they are identical.

Reviewers' comments:

Reviewer's Responses to Questions

**Comments to the Author**

1. Is the manuscript technically sound, and do the data support the conclusions?

Reviewer #1: No

Reviewer #2: Yes

Reviewer #3: Yes

2. Has the statistical analysis been performed appropriately and rigorously? 

Reviewer #1: No

Reviewer #2: Yes

Reviewer #3: N/A

3. Have the authors made all data underlying the findings in their manuscript fully available?

Reviewer #1: Yes

Reviewer #2: Yes

Reviewer #3: Yes

4. Is the manuscript presented in an intelligible fashion and written in standard English?

Reviewer #1: No

Reviewer #2: Yes

Reviewer #3: Yes

5. Review Comments to the Author

Reviewer #1: Although the research could potentially fill an existing knowledge gap, the provided data do not support the conclusion. First of all, the distribution of the stroke risk factors in case group and control group is not equal. Adjusting the stated risk factors would not solve the problem of confounding, as the potential confounders are more than those suggested by authors. Secondly and related to the first point, the risk of confounding by the indication is likely for observed results. Many details that could clarify the issue are missing, including the type of used NSAIDS, the dose (not only the duration) of used NSAIDs, the time between NSAID consumption and incidence of stroke, and so on. Moreover, for those who experienced stroke in the case group (and majority fell in the group of using more than 24 days/month) the cumulative used NSAID is unclear. The authors also do not provide any clue about the type of observed stroke in the cases and controls, whereby the plausibility of the conclusion could be improved. Another problem that may violate the conclusion of the study is the access to OTC NSAIDs which is totally neglected.

Since there are large scale meta-analyses that concluded no association between NSAIDs and stroke, in general, the authors could argue those findings only and only with providing the detailed information above to be able to reject the null hypothesis with a low risk of type one error.

The statistical analysis is not appropriately done. The used statistical software is not stated, as well as the specific statistical tests the authors applied to the data. Where overall means or proportions could be simply compared, those comparisons are missing.

The English should be improved, as there are a lot of grammatical defects.

Reviewer #2: In the selection criteria, The author did not mention the causes of dysmenorrhea on the basis of which they conducted this study.

The Authors didn't mention about how long the patients used the NSAIDs? How does taking NSAIDs twice can cause a stroke?

Reviewer #3: Good article. Consider to reword some of the discussion to ensure the audience able to follow. Grammar and typos need correction. Double brackets are confusing so please change. Ensure that the conclusion in abstract tally with the discussion overall. However, authors need to highlight the main findings for the study.

1. Good article with a lot of data presented in the manuscript.

2. Abstract: results section: statement is too generalised. Suggest to reword to highlight the important results

3. Abstract: conclusion does not reflect the objective of the study, consider to emphasize the main ideas

4. Page 8 : Statement ‘ the study results indicated…..’ suggest to put in the discussion

5. Figure 1 comparing 70(0.32%) controls vs 27 (0.83%) cases. Please indicate the p value

6. The authors suggest that NSAIDs have a high risk for stroke. However, the table 1 showed s those on NSAIDs also have other comorbidities (diabetes, hypertension etc) which are the traditional risk factors for stroke. How do the authors taken this into account? Arrhythmia, thyroid disease, hyperlipidemia accounted for the high number of cases. However, need to specify if it was atrial fibrillation as this will cause high incidence of embolic stroke

7. Authors did not specify what type of stroke in the article? Infarct or haemorrhage ? How was the diagnosis of stroke made?

8. In the text, certain words like ‘women with dysmenorrhea’ are in upper case. Please change to lower case. Double brackets in the results are confusing, suggest to change,

9. Discussion should highlight on the study’s main findings regarding NSAIDs use and stroke risk

10. Table 2 need to simplify to allow easy understanding

11. Table 3 – less than 5, use the symbol

12. References are not standardised. Books and journals, 7,8,17,20,21,23,24,25,27,28,30,31

6. PLOS authors have the option to publish the peer review history of their article (what does this mean?). If published, this will include your full peer review and any attached files.

Reviewer #1: **Yes: **Fazlollah Keshavarzi

Reviewer #2: No

Reviewer #3: No

---

## [Author Response · Author response to Decision Letter 0]

5 Oct 2021

Dear Dr.Muhammad Junaid Farrukh

Thank you for the comments and suggestion. We have revised the manuscript to ensure that our manuscript meets the publication style of PLOS ONE. 

(Please see the revised manuscript.)

---

## [Editor Report · Decision Letter 1]

12 Oct 2021

Stroke Risk associated with NSAIDs Uses in Women with Dysmenorrhea;a population-based Cohort Study

PONE-D-21-13819R1

Dear Dr. Ming-Hung Lin

We’re pleased to inform you that your manuscript has been judged scientifically suitable for publication and will be formally accepted for publication once it meets all outstanding technical requirements.

Kind regards,

Muhammad Junaid Farrukh

Academic Editor

PLOS ONE
---

## [Editor Report · Acceptance letter]

3 Nov 2021

PONE-D-21-13819R1 

Stroke Risk associated with NSAIDs Uses in Women with Dysmenorrhea : a population-based Cohort Study 

Dear Dr. Lin:

I'm pleased to inform you that your manuscript has been deemed suitable for publication in PLOS ONE. Congratulations! Your manuscript is now with our production department. 

Kind regards, 

on behalf of

Dr. Muhammad Junaid Farrukh 

Academic Editor

PLOS ONE